# Antibody-Conjugated Magnetic Beads for Sperm Sexing Using a Multi-Wall Carbon Nanotube Microfluidic Device

**DOI:** 10.3390/mi13030426

**Published:** 2022-03-10

**Authors:** Chalinee Phiphattanaphiphop, Komgrit Leksakul, Thananut Wanta, Trisadee Khamlor, Rungrueang Phattanakun

**Affiliations:** 1Industrial Engineering Department, Faculty of Engineering, Chiang Mai University, Chiang Mai 50200, Thailand; rug_chalinee@hotmail.com (C.P.); th.wata@gmail.com (T.W.); 2Animal and Aquatic Science Department, Faculty of Agriculture, Chiang Mai University, Chiang Mai 50200, Thailand; trisadee.kha@gmail.com; 3Synchrotron Light Research Institute, Nakhon Ratchasima 30000, Thailand; rungrueang@slri.or.th

**Keywords:** monoclonal antibody conjugated, sperm sexing, multi-wall carbon nanotubes, magnetic beads, microfluidic

## Abstract

This study proposes a microfluidic device used for X-/Y-sperm separation based on monoclonal antibody-conjugated magnetic beads, which become positively charged in the flow system. Y-sperms were selectively captured via a monoclonal antibody and transferred onto the microfluidic device and were discarded, so that X-sperms can be isolated and commercially exploited for fertilization demands of female cattle in dairy industry. Therefore, the research team used monoclonal antibody-conjugated magnetic beads to increase the force that causes the Y-sperm to be pulled out of the system, leaving only the X-sperm for further use. The experimental design was divided into the following: Model 1, the microfluid system for sorting positive magnetic beads, which yielded 100% separation; Model 2, the sorting of monoclonal antibody-conjugated magnetic beads in the fluid system, yielding 98.84% microcirculation; Model 3, the sorting of monoclonal antibody-conjugated magnetic beads with sperm in the microfluid system, yielding 80.12% microcirculation. Moreover, the fabrication microfluidic system had thin film electrodes created via UV lithography and MWCNTs electrode structure capable of erecting an electrode wall 1500 µm above the floor with a flow channel width of only 100 µm. The system was tested using a constant flow rate of 2 µL/min and X-/Y-sperm were separated using carbon nanotube electrodes at 2.5 V. The structure created with the use of vertical electrodes and monoclonal antibody-conjugated magnetic beads technique produced a higher effective rejection effect and was able to remove a large number of unwanted sperm from the system with 80.12% efficiency.

## 1. Introduction

Bovine sperm separation technology has had a significant impact on both the food and livestock industries, leading to an increase in the demand for assisted reproductive technologies (ARTs). Female cattle are required for the dairy industry, while male cattle are preferred in the beef industry [1] since male cattle have a high daily growth rate and can be fattened easier than female cattle. With a value estimated at more billion dollars, the knowhow and technology provide specific solution for improving their breeding. In order to identify the type of sperm cell in semen to deliver the most possibility of sperm sexing in agricultural and farm system, many parameters of sperm have been studied based on their characteristic of X and Y chromosome. Not only the size, mobility, and DNA content of X and Y chromosomes are different, but also the electric charge or zeta potential on their surface is slight difference [2]. The X-sperm has a greater net negative charge (−20 mV) than Y-sperm (−16 mV) [3,4], which affects their mobility under electrostatic field of 1.07 and 0.41 μm s^−^^1^ V^−^^1^ for X and Y chromosome-containing spermatozoa, respectively. Furthermore, male and female mammalian sperm cells differ in sialic acid content. It was shown that the motility of X and Y chromosome-containing sperm cells differs under a magnetic field, where the female sperm cell is faster than the male.

To deploy the difference characteristic of X and Y chromosome for sperm sexing, many sorting techniques have been proposed to achieve high efficiency. The conventional centrifugation based on swim-up (CSW) and direct swim-up (DSW) play roles in non-complicated method but comprise various centrifugation steps and oxidative stress, resulting in sperm DNA damage. Sperm sexing methods based on monoclonal antibody (MAb) specific to Y chromosome-bearing sperm have been used to reduce the sex ratio of male calves. Although the preliminary laboratory tests revealed a higher ratio of X-chromosome bearing sperms than that of Y-chromosome bearing sperms [5], MAb has some limitations in variability, static, and unevenness. The conventional method for Y-sperm trapping was processed in the test tube in which a magnet was used to trap the magnetic beads at the bottom of the tube and allowed X sperm swim up, followed by sucking them out with micropipette. However, this method could not deliver all X-sperms from the test tube [6]. Nowadays, flow cytometry is considered the most reliable and effective method to obtain spermatozoa of the desired sex with higher than 90% purity [7,8]. Regarding the properties of sperms, X-sperms have 3.8% more DNA content than Y-sperms, which provides an opportunity to separate them based on DNA content. The disadvantages of using flow cytometry in sexing sperm are as follows: (1) sexing rate is quite low at approximately 10 × 10^6^ sperms/h, as 20 × 10^6^ bull sperms are required for each artificial insemination, resulting in unfeasible sexing lead time during artificial insemination; (2) flow cytometers are expensive (250,000 USD per device); (3) the conception rate has reduced from 50% to 42%; (4) flow cytometry requires a trained technician [9]. In the past 10 years, lab on a chip (LOC) has supported sperm analysis and sorting discussed by Samuel et al. [10]. By designing of micropatterning of microfluidic channel, LOC integrated with sensors and actuators are employed such as fluorescence-activated cell sorting (FACS) [11,12], dielectrophoretic sorting [13], electrophoretic sorting [14,15,16], magnetic-activated sperm cell sorting [17,18,19], sperm sorting to utilize the difference in motility or nonmotility [20,21,22,23,24,25], immunological markers [26,27], electrokinetic isolation [28], inertial separation [29], and controlled pressure sorting [30]. Another simple technique which improved to acquire higher effective by optimizing microfluidic chip parameters based on electrophoresis was proposed with successful sorting up to an accuracy of 70% [16]. The microfluidic chip utilized multiwall carbon nanotube (MWCNT) thick electrode to expand the vertical side wall area, resulting in high density of electrostatic field which affected to the sperm mobility based on the net negative charge of X and Y sperms in microfluidic system.

To increase the performance of sperm sorting based on electrophoresis technique, we present LOC integrated with MWCNT electrodes, which provides the possibility for increasing the performance both effectiveness and productivity. The conventional method of magnetic beads trapped the Y-sperms in a test tube.

Based on the property of Mab, which traps the Y-sperms and conjugates with magnetic beads directly, electrophoresis microfluidic chip can be utilized to control the moving direction of magnetic beads easily, resulting in Y-sperms sorting from X-sperms. Based on the concept of MAb conjugated magnetic beads working under MWCNTs electrodes in microfluidic chip, all sperms will interact under the same electrostatic filed condition, especially for magnetic beads, which offer positive charge on the surface that will be attracted to the negative electrode. The Y-sperms trapped to magnetic beads were sorted out at the exist microchannel, which allowed the X-sperms to flow. Moreover, this strategy can apply a permanent magnet to keep all magnetic beads at the exist microchannel to collect the X-sperms.

In this study, antibody-conjugated magnetic beads for sperm sexing using a MWCNTs microfluidic device is proposed to increase the efficiency of electrophoresis sperm sorting technique. Magnetic beads with positive charge attached to Y-sperms with MAb and moved them to the MWCNT negative electrode in microfluidic chip. Most of magnetic beads were forced to move to the exist microchannel leaving all X-sperms to interact with the electric field, thereby purifying them. In addition, attracting magnetic beads with a permanent magnet at the outlet of the microfluidic chip can improve the quality of the desired sex sperm and show specificity and accuracy in terms of gender separation.

## 2. Materials and Methods

### 2.1. Conceptual Design for Novel Sperm Sorting Method

The conventional magnetic beads applied for sperm sorting [6] is shown in Figure 1a. For better efficiency and high X-sperm collection yield, a microfluidic system was applied with magnetic beads as shown in Figure 1b. The semen was bound with MAb conjugated magnetic beads in a test tube and fed into a microfluidic chip. MAb conjugated with positive charge magnetic beads can captured Y-sperms and moved them to a negative electrode easily, leaving the X-sperms in the main channel. For separation, the laminar flow in the microchannel moves the X-sperms out of the magnetic beads to a positive electrode, which guarantees that all X-sperm were processed in the same condition. Meanwhile, the Y-sperm separation using magnetic beads could not offer high rate or separation since the MAb magnetic beads can trap the Y-sperms at 74.46% efficacy. Therefore, the electric charge on the sperms’ surface of both X and Y were preliminary sorted under electrostatic field at 74% successful rate [16] and delivered into exist microchannels before being trapped with magnetic beads, resulting in high rate of X-sperm separation. To avoid the flow resistance problem in the microchannels due to the accumulated of magnetic beads, both output samples were delivered directly to the outside test tubes after separating the magnetic beads using a permanent magnet at the bottom of the test tube as shown in Figure 1b. In this study, the Y-sperm sorting based on MAb conjugated magnetic beads without any permanent magnet trapping is first tested at the exist microchannel, after which the efficiency of Y-sperm sorting is performed.

### 2.2. CNTs (MWCNTs)

Carbon plays an important role in the development of electrochemical biosensors [31,32,33] and is used as a promising material for sensing applications in a wide range of electrical, chemical, mechanical, and structural property fields [34,35,36]. Therefore, MWCNTs were chosen for the sperm isolation test. The MWCNTs were tested with a four-point probe for conductivity data, with the results listed in Table 1, and the MWCNs material properties were examined based on scanning electron micrographs as seen in Figure 2.

### 2.3. Magnetic Beads

Magnetic beads were used to add charge to the sperm with monoclonal antibody 1F9, since monoclonal antibody 1F9 is specific to Y sperm. According to the hypothesis of different surface charges of X and Y sperms, the X sperms were had a greater net negative charge than Y sperms [3] The applied magnetics particle beads (Pierce™ NHS-Activated Magnetic Beads; Thermo Scientific™, Life Technologies Holdings Pte Ltd., 33 Marsiling Industrial Estate Rd 3, Singapore) were used for the preparation of MAbs conjugated with magnetic particles. The magnetics particle beads have a mean diameter of magnetic particles of 1 µm (nominal), the density of magnetic particles was 2.0 g/cm^3^ with a binding capacity of ≥ 26 µg of rabbit IgG/mg of beads. The magnetic beads have an amine functional group consisting of an N atom bonded either to C or H atoms via σ bonds. Both the C-N and N-H bonds are polar due to the electronegativity of the N atom. Therefore, these magnetic beads have positive charges that associate with the Y sperm, the unwanted target sperm, and are isolated using a fabricated microfluidic structure to remove them, leaving only the X sperm. To ensure sperm X and Y purity, those that could still be aspirated by the microfluidic structure were re-purified via magnetic traction. This ensures that the final X sperm have the highest separation rate and are not contaminated with any material.

### 2.4. Preparation of Monoclonal Antibodies Conjugated with Sperm

#### 2.4.1. Control Sample

The MAb reaction test conducted using a flow cytometer for screening of sperm sex was divided into two groups: the control sample test and its negative result (control and negative control, respectively) and a monoclonal antibody 1F9, which is an IgG antibody that binds to the H-Y antigen located on the surface of Y sperm-specific cells. The control samples were A01 and A02; A01 was the conjugate control (CC), which consisted of 50 μL of FACS buffer and 50 μL of bull sperm; A02 was isotype control (IC), which consisted of 1 μL of ST28A antibody, 50 μL of FACS buffer, and 50 μL of bull sperm. However, there is no specificity for the sperm that it should not bind to the sperm. The control samples A01 and A02 are collectively referred to as negative controls (NC).

#### 2.4.2. Monoclonal Antibody 1F9

Comparison with the NC sample revealed that the MAb 1F9 affected the bindable sperm, as mentioned above. NCs are unable to bind to sperm. However, the positive results obtained with the 1F9 MAb revealed that the 1F9 MAb could bind to sperm. A03 consisted of 50 μL of 100 μg/mL MAb, with a final 1F9 MAb concentration of 50 μg/mL. A04 consisted of 50 μL of 50 μg/mL MAb and 50 μL of bull sperm, with a final 1F9 MAb concentration of 25 μg/mL. A05 consisted of 50 µL of 25 µg/mL MAb and 50 µL cow sperm, with a final 1F9 MAb concentration of 12.5 µg/mL. Finally, A06 consisted of 50 µL of 12.5 µg/mL MAb and 50 µL of cow sperm, with a final 1F9 MAb concentration of 6.25 µg/mL. The parameter of test sample A01-A06 was describing at the Table 2.

### 2.5. Preparation of Monoclonal Antibodies Conjugated with Magnetic Particles

Magnetic particles were prepared by balancing protein and magnetic particles at 25 °C, and then 300 μL of magnetic particles were inserted into a 1.5 mL microcentrifuge tube at shown the procedure in Figure 3. Next, the tubes were placed in a magnetic stand, magnetic particles were collected, and suspended solids were disposed. Ice-cold 1 mM hydrochloric acid (1 mL) was added to the tube and blended for 15 s. The tubes were placed in a magnetic stand, magnetic particles were collected, and suspended solids were discarded. Then, 300 μL of protein solution (1 mg/mL in coupling buffer) was added to the tube and blended for 30 s. The tube was incubated for 2 h at room temperature on a rotary agitator. During the first 30 min of incubation, the mixture was mixed for 15 s every 5 min and then every 15 s every 15 min, until the curing was complete. The tube was then placed in the magnetic stand, the magnetic particles were collected, and the suspended solids were discarded. Then, 1 mL of 0.1 M glycine (pH 2) was added to the magnetic particles and mixed for 15 s, and the tubes were placed in the magnetic stand. The magnetic particles were collected, and the solution of suspended solids was discarded; this process was repeated once. Thereafter, 1 mL of ultrapure water was added to the magnetic particles and mixed for 15 s in the magnetic stand. The magnetic particles were collected, while the suspended solids were discarded. A 1 mL of 3 M ethanolamine (pH 9) was added to the magnetic particles, blended for 30 s, and incubated for 2 h at room temperature on a rotary agitator, and the tubes were placed in a magnetic stand. The magnetic particles were collected and the suspensions were discarded again. Subsequently, 1 mL of purified water was added and mixed well, magnetic particles were collected with a magnetic stand, and suspended solids were discarded before adding 1 mL of coupling buffer (50 M borate, pH 8.5) with 0.05% sodium azide and mixed well. The magnetic particles were collected with a magnetic stand, discarding the suspension, and the process was repeated two more times, with 300 μL of coupling buffer and 0.05% sodium azide, mixed well, and stored at 4 °C until use. Finally, MAbs conjugated with magnetic particles were rechecked through flow cytometry. Then, A07 and A08 were prepared and experimented. A07 was negative control (NC), which consisted of 2.5 μL of 1F9 and 50 μL of PBS. A08 was working solution (WS), which consisted of 2.5 μL of 1F9, 50 μL of PBS and 5 μL of Gαm-FITC (1:100).

### 2.6. Monoclonal Antibodies Conjugated with Magnetic Particles and Sperm

The Holstein Friesian bull semen samples used in this study were purchased from the Dairy Farming Promotion Organization of Thailand. A single straw of bull semen sample containing 30 × 10^6^ spermatozoa preserved in liquid nitrogen was thawed in a water bath at 37 °C for 40 s. Thereafter, the sample was placed into a 1.5 mL microcentrifuge tube and stored in a chamber at a temperature of 37 °C. Live sperm with mortality greater than 70% was used. One microliter of warm PBS extender at 37 °C was added to remove the egg yolk extender from the spermatozoa. Subsequently, a tube of sperm sample was centrifuged at 16,128 g for 10 s thrice, and the supernatant was discarded to collect the sperm pellet. Turk’s solution was mixed with sperm at a ratio of 1:10. Then, the sperm was blocked with AB serum (FC Receptor) for 30 min. Finally, a total of six sperm tubes were prepared: Tube 1, monoclonal antibody 100 μg/mL, 50 μL volume, 50 μL sperm, and A08. Tube 2, 50 μg/mL monoclonal antibody, 50 μL volume, 50 μL sperm, and A08. Tube 3, 25 μg/mL monoclonal antibody, 50 μL and 50 μL sperm and A08. Tube 4, 12.5 μg/mL monoclonal antibody, 50 μL volume, 50 μL sperm, and A08. Tube 5, 1 μL ST28A antibody, 50 μL FACS buffer, 50 μL sperm, and A08. Tube 6, 50 μL FACS Buffer, and 50 μL sperm and A08. The best-conjugated combination was in Tube 3.

### 2.7. Microfluidic Device Fabrication

#### 2.7.1. Fabrication of Microfluidic Chip for Thin Film Electrode

The design of the microfluidic chip, which mainly consists of a Y channel cell microfluidic channel with a pair of electrodes positioned at the separation regions, is shown in Figure 1b. The microfluidic chip was designed using a layout editor program. The dimensions of the microfluidic chip were 6000 µm × 150 µm, with a thickness of 50 µm. A pair of Ti/Au electrodes was deposited with dimensions of 5000 µm × 500 µm and a thickness of 100 nm.

The microfluidic chip was placed on a glass wafer using UV and soft lithography processes. Figure 4 illustrates the microfabrication process. To create a microelectrode, thin film of Ti/Au were deposited via a thermal evaporator, following by coating of AZ1512 photoresist for UV lithography patterning. The exposed areas were developed to reveal the metal surfaces which were removed by wet etchant of 1 (HCI): 3 (HNO_3_) and HF3% for Au and Ti layer, respectively. Photoresist, last, was removed with acetone to achieve the final electrodes (Figure 4a). To create a microchannel, the SU-8 photoresist (SU-8 dry-sheet 50 µm) for the microstructure was laminate-coated and patterned. UV exposure was performed on SU-8 at 60 s to achieve a thickness of 50 µm of SU-8. The entrapping structure was designed to be wider than the space between the two sidewall electrodes to create the detection signal in the chamber. To provide the inlet and outlet flow of fluidic interconnections, a replicated polydimethylsiloxane sylgard 184 (PDMS-184) was used as the master mold replication (Figure 4b). Finally, PDMS which have a microchannel-patterned underneath, were bonded by the plasma O_2_ method on a pair of Ti/Au electrodes (Figure 4c).

To set up the electrical base on the microelectrode connection, the microdevice was attached to the print circuit board and connected by using the wire bonding method in the experimentation.

#### 2.7.2. Fabrication of Microfluidic Chip for MWCNT Electrode

To improve the effectiveness of sperm cell separation, MWCNTs were added to the microfluidic chip for enhancing high electrical field concentration. The microfluidic chip with MWCNTs for X and Y sperm separation was fabricated via microfabrication technology using UV and X-ray lithography processes as shown in Figure 5.

The area of the microelectrode was created through Ti/Au evaporation process. The dimensions of MWCNTs area placed on the microelectrode were 5000 μm ×100 μm, with a thickness and width of mainstream of 1500 μm and 100 μm, respectively. The microelectrode was patterned through UV lithography and wet etching on a glass substrate as shown in Figure 5a. PDMS microchannel was duplicated from a photoresist structure via X-ray lithography process. Since the PDMS channel has high thickness and UV- lithography does not yield sharp wall-angles in the structure, the UV-lithography was replaced by X-ray lithography. MWCNTs were then created as electrodes with a hydraulic machine and placed into replicated PDMS, and then on the Ti/Au microelectrodes as shown in Figure 5b. Finally, the MWCNTs electrodes microfluidic chip on a glass substrate was ready for experimentation as shown in Figure 5c. The sperms were then loaded into the chip and analyzed through real-time PCR.

## 3. Results

### 3.1. Testing of Monoclonal Antibodies Conjugated with Sperm

The controls for the MAb conjugated with sperm, A01 and A02, were analyzed using flow cytometry. The controls were found to be negative, as shown in Figure 6a,b. The samples containing MAb 1F9 combined with a sperm-binding Y sperm were found to be positive, as shown in Figure 6c–f. The MAb was conjugated with sperm at various concentrations, and yielded different results with the sperm test. The sperm test with the ST28A antibody yielded a negative result, showing that the sperm and the 1F9 MAb bind together at reduced concentrations. The sample with optimum concentration of A04 for sperm binding was 25 μg/mL (1F9 MAb 50 µL with 25 μg/mL concentration and sperm 50 µL).

### 3.2. Testing of Monoclonal Antibodies Conjugated with Magnetic Beads

The result of the flow cytometry test of the A07 sample was negative. The result of the A07 sample was used as a NC sample for comparison with the A08 sample. The A08 sample consisted of a GαM-FITC-dyed magnetic bead on the MAb, as shown in Figure 7. The result of the flow cytometry test of the A08 sample was positive and when compared with the negative result of the A07 sample, the magnetic bead and MAb were found to be compatibility each other. The compatibility was 94.54%, as shown in Table 3.

### 3.3. Testing the Magnetic Particle Beads in a Microfluidic Chip

The magnetic particle beads were separated using a microfluidic chip. The properties of the magnetic beads were attributed to the positively charged N-hydroxysuccinimide (NHS) functional groups on the blocked magnetic bead surface. Using electrophoresis, the magnetic beads were tested by separating the beads on a microfluidic device using the principle of the charge on the surface of magnetic beads. Electrophoresis is a laboratory technique used extensively to separate DNA, RNA, or protein molecules based on their size and electrical charge. Therefore, the optimized condition for the separation of charged beads, as understood from Phiphattanaphiphop’s research, included a microfluidic device with a 150 μm microchannel and 100 μm electrode distance and showed the highest performance of 87.07% validation (95% confidence level) for the separation of negatively charged TiO_2_-coated PS beads [19]. An optimally designed structure used previously for the separation of the microfluidic beads was used to test the magnetic particle beads. The magnetic particle beads were prepared by controlling the sample flow rate at 2 µL/min. This microfluidic system device was able to separate magnetic particle beads (Figure 8).

Moreover, magnetic particle beads were tested on the MWCNT electrode microfluidic chip. It was processed into a pelletized form for use in the microelectrode area. The magnetic particle beads were controlled using a BS-8000 syringe pump (Braintree Scientific, Inc., Braintree, MA, USA) at a flow rate of 2 µL/min. The MWCNT-microfluidic system device could separate magnetic particle beads with 100% success, as shown in Table 4. The separation efficiencies of magnetic beads were calculated using a hemocytometer plate and the number of magnetic beads were counted under a microscope.

### 3.4. Testing the Monoclonal Antibodies Conjugated with Magnetic Beads in a Microfluidic Chip

The MAb conjugated with magnetic particles for better sperm separation was pumped into the microfluidic chip. An experiment was conducted for separating various particles using magnetic particles on a MAb based on the properties of MAbs and magnetic particles by controlling the sample flow rate at 2 µL/min. It The microfluidic device could separate the magnetic particles on the MAb. The percent of sample successfully sorted results were achieved at 95.42%, as shown in the Figure 9. Moreover, when MAb conjugated with magnetic particles was again conducted with MWCNT-microfluidic chip, the results were in line with the theory, that is, the rate of separation of MAb conjugated with magnetic particles with higher positive charge at 98.84%, as shown in Table 5. The efficiencies of monoclonal antibodies conjugated with magnetic beads were calculated using a hemocytometer plate and the number of magnetic beads were counted under microscope.

### 3.5. Testing Sperm and Sperm with Monoclonal Antibodies Conjugated Magnetic Particle Beads in Microfluidic Chip

Sperm and sperm with MAb-conjugated magnetic particle beads were added to achieve better separation on a microfluidic chip compared with that on a MWCNT-microfluidic chip. The sperm with MAbs conjugated with magnetic particles in a microchannel in a microfluidic system device and microelectrode was tested. The particle flow was observed in the microscopic flow channel. The particle inflow rate was controlled at 2 µL/min, and the electric fields were discharged at 2.5 V for the microfluidic chip. The results analyzed using RT-PCR were achieved at 61.11% and 80.12% for sperm conjugated MAbs and magnetic particle beads, respectively, as shown in the microfluidic chip and MWCNT-microfluidic chip, respectively in Figure 10. The results of the sperm in combination with MAbs conjugated with magnetic particles revealed better efficacy compared with those by non-antibody coadministration with beads, as shown in Table 6.

## 4. Discussion

The micro-fluid system and micro-electrodes were created for separating magnetic particles beads, monoclonal antibodies conjugated with magnetic particle beads and sperm and sperm with monoclonal antibodies conjugated magnetic particle beads with the device. The particle flow was observed in the microscopic flow channel. In the present study, three sorting patterns were examined: Model 1, the microfluid system for sorting positive magnetic beads; Model 2, the sorting of monoclonal antibody-conjugated magnetic beads in the fluid system, and Model 3 sorting monoclonal antibody-conjugated magnetic beads with sperms in the microfluid system. Two types of separation structures were designed; the first was designed by generating a thin-film electrode with a thickness of 50 nm via the UV lithography process. The second was designed by generating a wall electrode with a larger thickness that generated an electric field using MWCNT to obtain a vertical wall structure with a thickness of 1500 µm. Owing to the thickness of the electrode in the second design and as UV-lithography does not yield sharp wall angles in the structures, X-ray lithography was used instead of UV-lithography. As shown in Figure 5, it is considered a vertical electrode fabrication technique that performs better than thin-field electrode formation technique and is more efficient in separating charged particles. By controlling the inflow rate of particles and emitting electric fields of various sizes, the following results were obtained.

### 4.1. Magnetic Particles Beads in Microfluidic System

The results of the magnetic particle separation experiment in which the inflow rate of magnetic particles was maintained at 2 μL/min, with a 2.5 V electric field, revealed that the magnetic particles were released into the microstructure flow channel in the microfluidic chip. Magnetic particles beads flowed through the main flow channel and into the negative-ion flow channel. The magnetic particles released into the microstructure flow channel were positively charged at the north poles. Therefore, the magnetic particles flowed out of the negative-ion flow channel rather than the positive charge flow channel. According to the theory of electricity and magnetism, particles with same polarity or charge repel each other. This shows that the generated microfluid system device can separate magnetic particles with a success rate of 100% on both the microfluidic chip and MWCNT-microfluidic chip, as shown by flow cytometry.

### 4.2. Monoclonal Antibodies Conjugated with Magnetic Particle Beads

The MAb 1F9 was conjugated with magnetic particle beads. MAb 1F9 only reacts with Y sperms; in other words, MAb 1F9 binds to Y sperms. Positively charged magnetic particle beads conjugated with MAbs 1F9 are attracted by the generated microfluidic device structure. By generating an electric field at 2.5 V, the device could selectively screen antibody-coated magnetic beads with a success rate of 95.42% in thin-film electrodes microfluidic chip and 98.84% in the MWCNT electrodes microfluidic chip. The analyses were conducted using a flow cytometer. Therefore, the magnetic particles flowed out of the negative electrode flow channel rather than the positive electrode flow channel. The MAbs with magnetic particles had high effect on electric field, which could adjust the amplitude voltage from 1 to 2.5 V. The magnitude of the electric field at 2.5 V was the maximum value of the electric field, which can be used without forming air bubbles or adversely affecting the microfluidic systems. This shows that the generated microfluidic system device can isolate magnetic particle beads on a MAb.

### 4.3. Sperm and Sperm with Monoclonal Antibodies Conjugated Magnetic Particle Beads

Sperm sexing experiments conducted on magnetic particle beads on a MAb via control of the inflow rate of the sample at 2 μL/min and release of a 2.5-V electric field revealed that the sperms working with magnetic particles on the MAb were released into the microstructure flow channel. Sperms with monoclonal antibodies conjugated magnetic particle beads flow through the main flow channel separately to both exits—the positive charge flow channel and the negative charge flow channel. To obtain clear results, the accuracy was checked using real-time PCR.

The analysis of the sperm screened with the microfluidic system using real-time PCR was used to determine the DNA content of the Y sperm in the sample. From the conventional method of thin film electrodes and MWCNTs electrodes microfluidic chip [15] which does not utilize MAb conjugated magnetic beads, the results of Y-sperm separation were 53.99% and 74.62%, respectively. For this experiment, the sperms with MAbs at the magnetic bead conjugate were isolated at 61.11% in the thin film electrodes microfluidic chip and at 80.12% in the MWCNT electrodes microfluidic chip. It can be seen that the MAbs with the magnetic bead conjugate result in superior sperm separation in both the thin-film electrodes microfluidic and MWCNT electrodes microfluidic chips. Based on the principle of separation using a MAb and magnetic beads to increase the screening capacity of sperm using an electric field, the result supports the hypothesis that an improved non-MAb in the magnetic bead conjugate could be achieved with a success rate of up to 80%. The X-sperm motility under electric filed is not affected by magnetic beads and delivered the same results, while Y-sperm results could be improved under electric field at the mainstream without applying magnetic field at the exist microchannel. However, the magnetic bead conjugate MAb method can only be used for specific samples. Specifically tailored to specific sex needs, samples with conjugated antibodies were not available, and the sample used for this study was sperm. The size of the X sperm is 20% of the size of Y sperm, and has greater ratio, which resulted in an increased success rate in obtaining X sperm.

Although 5% increase in Y-sperm separation efficiency was not enough for bovine sperm sexing process when compared with the conventional method, it can be improved by applying a permanent magnet at the exist of microchannel and adjust microchannel parameters. Only MAb conjugated magnetic beads showed an 80% success rate under electrophoresis, suggesting that the X-sperm success rate is also increase.

### 4.4. Magnetic Beads and Electrophoresis Sperm Separation

The new method for X-sperm sorting by trapping of Y-sperm with magnetic beads was proposed to achieve high efficiency when compared with the basic electrophoresis sorting method without magnetic beads. The process was based on electrophoresis, which utilized electric field to separate X-sperm in the mainstream and Y-sperm with MAb conjugated magnetic beads. From the experimental results, sorting method with magnetic beads provided a better efficiency than that without magnetic beads since positive charge magnetic beads can enhance Y-sperm sorting efficiency at the exist microchannel to 80%. Based on the same moving distance in the mainstream, the flow of X-sperms to a positive electrode was faster than that of Y-sperms since the net negative charge of X-sperm was greater than that of Y-sperm, resulting in the distribution of X-sperm at the upper exist microchannel by adjusting the applied electric field condition from 1 to 2.5 V. For the MWCNT electrodes microfluidic chip, moreover, the length of the mainstream for electrophoresis separation and height of the microchannel were 13,000 µm and 1500 µm, respectively, allowing X-sperms to be continuously attracted by a positive electrode along the flow stream and separating them from Y-sperms and Y-sperm conjugated magnetic beads at the exist microchannel. To achieve higher efficiency, microchannel parameter adjustment may be necessary, especially for flow resistance, microchannel dimension, and number of exist microchannels. Since there were three types of charge particles in the mainstream, Y-sperm conjugated magnetic beads, Y-sperm non-conjugated magnetic beads, and X-sperms, the exist microchannel could be designed for three corresponding microchannels by varying the width. This design would allow the charged particles to be directed to a specific fraction by adjusting the applied electric field [37] as well as a permanent magnet trapping integrated with the microfluidic chip. Furthermore, this method’s ability to isolate with good morphology and low levels of DNA fragmentation when compared with normal electrophoresis method should be considered, especially for magnetic field trapping at the exist of microchannel for the sperm survival rate and DNA damage after separation and purification.

## 5. Conclusions

In this study, magnetic particle beads were designed for conjugation with MAb-specific Y sperm to separate it from X sperm, with the objective of obtaining the X sperm for commercial use in dairy cow production. Separation of magnetic particles beads, monoclonal antibodies conjugated with magnetic particle beads and sperm, and sperm with monoclonal antibodies conjugated magnetic particle beads were performed using a microfluidic system, designed using MWCNTs for separation by sexing X and Y sperms. Separation was performed based on the assumption that the X and Y sperms had different electrical charges on the surface. The microfluidic devices used here were designed based on the optimal values from previous research in which a microfluidic device with a 150-μm microchannel and 100-μm electrode distance were used, and the structure was obtained from MWCNT-microfluidic chip [19]. Based on the sorting test results, in which MAbs were conjugated with the magnetic particle beads, the device could sort the magnetic particle beads, MAb-conjugated magnetic particle beads, and sperm-conjugated MAbs with success rates of 100%, 95.42%, and 61.11%, respectively, in the microfluidic structure which has a flat-plane electrode thickness of 50 nm created via UV lithography. The MWCNT-microfluidic chip successfully sorted magnetic particle beads, MAb-conjugated magnetic particle beads, and sperm-conjugated MAbs with magnetic particles, with success rates of 100%, 98.84%, and 80.12%, respectively. The MWCNT-microfluidic chip was designed to have an electrode wall that could increase the electric field, causing the desired sperm to be pulled out. The electrode wall was created by X-ray lithography with a thickness of 1500 µm, which effects pulling the conjugated magnetic particle beads, giving them the ability to pull out the Y sperm. This allows the system to take the desired X sperm for further fertilization. The original device sorted the antibody-free sperm in the microfluidic device at only 53.99% success rate, while in the MWCNT-microfluidic chip at 74.62% success rate; therefore, success in sperm sorting when the MAb is applied to the magnetic particle beads is higher than without it, but there are limitations in the specificity of the MAb application because it can result in only one gender segregation. Moreover, it can be seen that the best sperm-sexing structures are those created using MWCNTs in microfluidic devices. Sperm separation was performed using an electric field separation principle and a microfluidic structure designed using MWCNTs to optimize the separation and magnetic bead conductivity and achieve better rejection. This separation result supports the hypothesis that X and Y sperms have different electrical charges on their surfaces.

## Figures and Tables

**Figure 1 micromachines-13-00426-f001:**
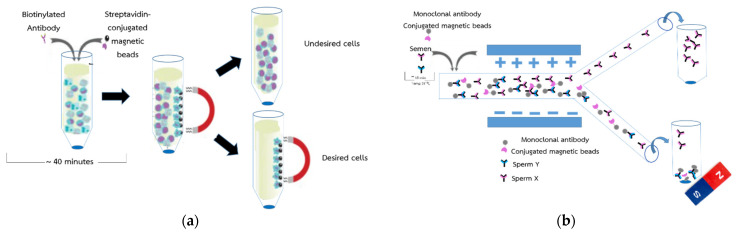
The conceptual of the sorting of sperm X and Y (**a**) shown the method for separation efficiency of protein by magnetic beads and (**b**) shown the method for separation efficiency of sperm by magnetic beads.

**Figure 2 micromachines-13-00426-f002:**
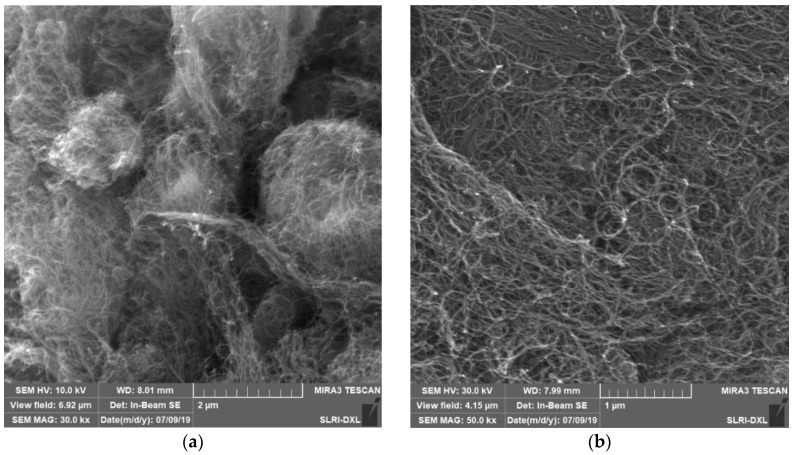
(**a**) Scanning electron micrograph of multiwalled carbon nanotubes powder at 1000× magnitude and (**b**) Scanning electron micrograph of multiwalled carbon nanotubes bulk at 3000× magnitude.

**Figure 3 micromachines-13-00426-f003:**
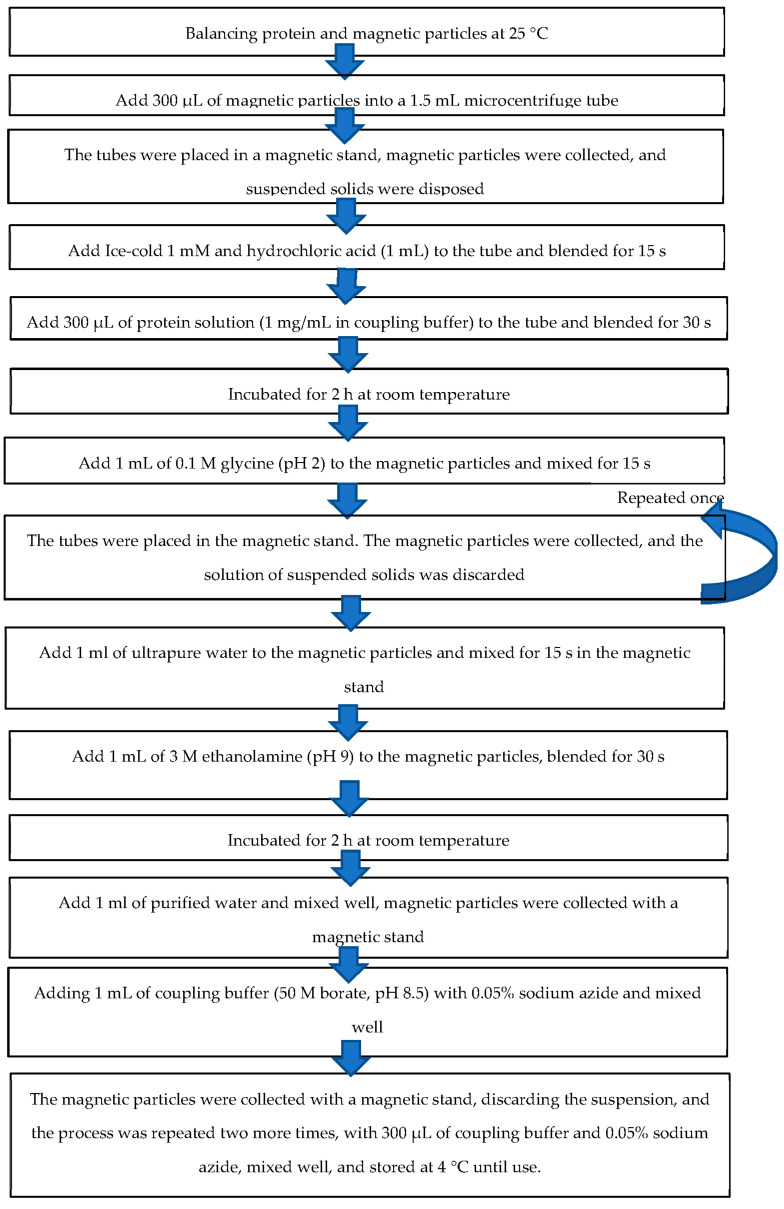
The procedure of preparation of monoclonal antibodies conjugated with magnetic particles.

**Figure 4 micromachines-13-00426-f004:**
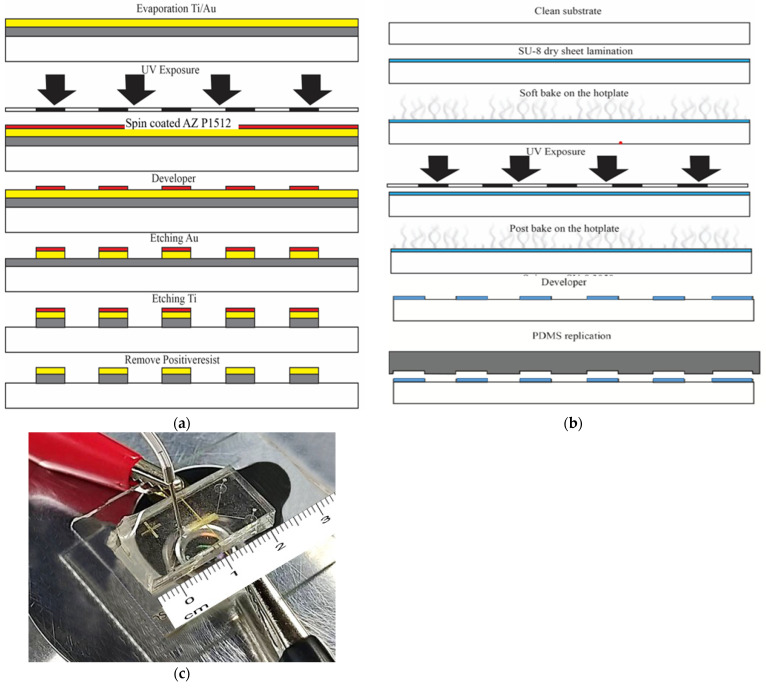
Illustration of the microfluidic device fabrication process using ultraviolet and soft lithography. (**a**) Fabrication of the microelectrodes (**b**) fabrication of the microchannels and (**c**) Microfluidic chip for thin film electrode.

**Figure 5 micromachines-13-00426-f005:**
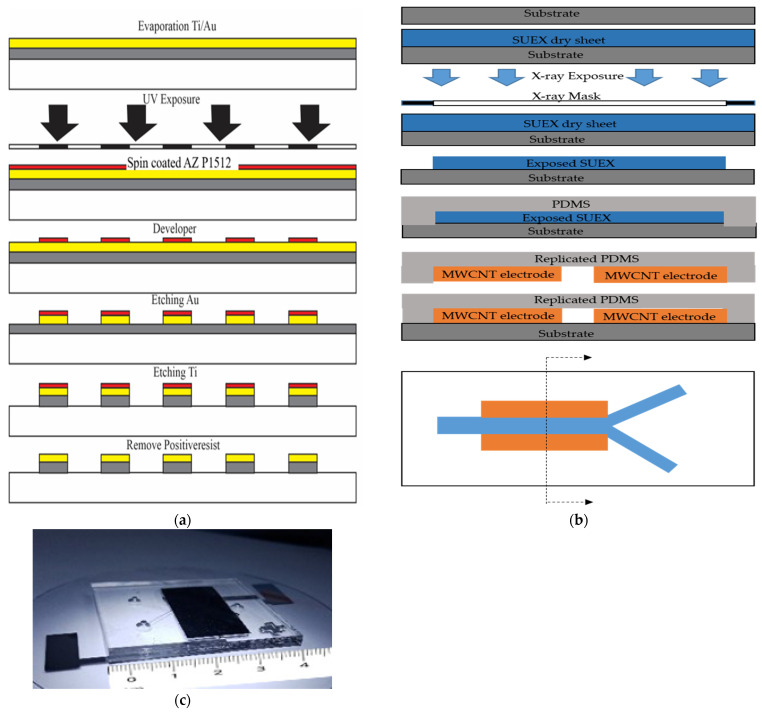
Illustration of the microfluidic device fabrication process using X-ray photolithography and soft lithography. (**a**) Fabrication of the microelectrodes (**b**) fabrication of the microchannels with MWCNT and (**c**) Microfluidic chip for MWCNT electrode.

**Figure 6 micromachines-13-00426-f006:**
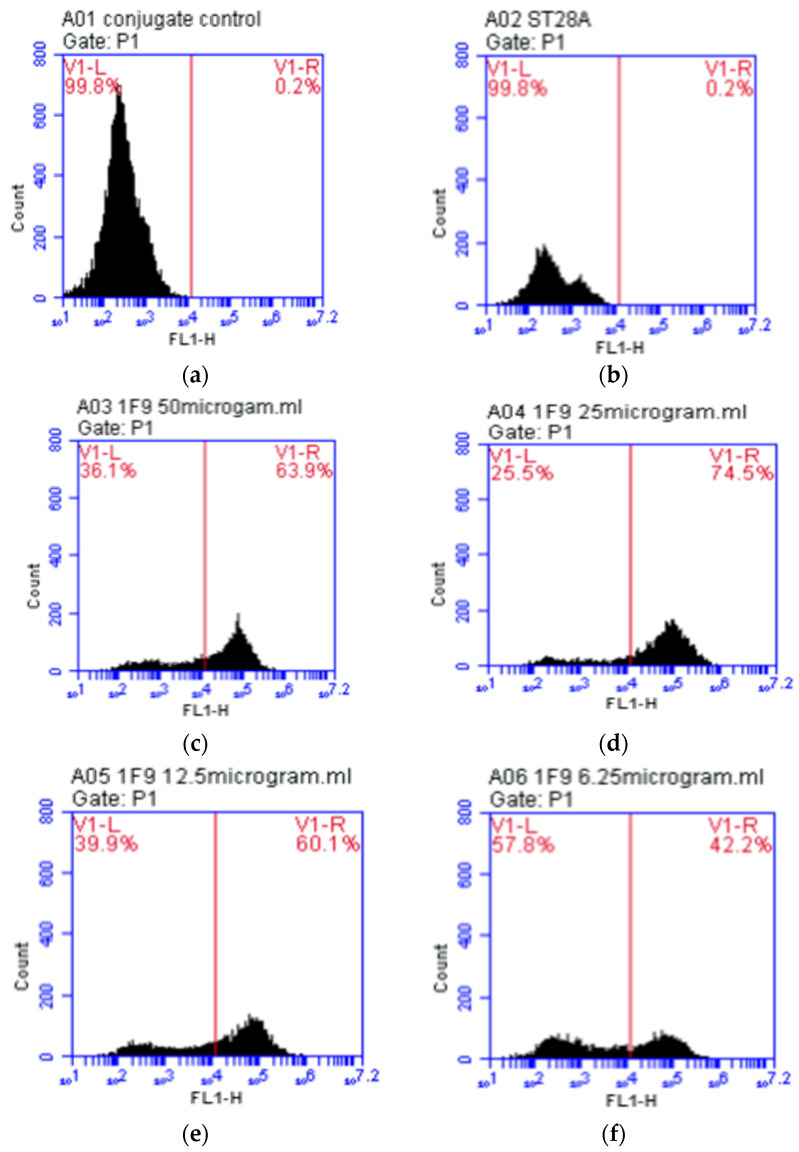
The relationship between the fluorescence concentration (FL1-H) and the number of cells (Count) (**a**) the control sample (**b**) the control sample ST28A (**c**) The sample A03 (**d**) The sample A04 (**e**) The sample A05 and (**f**) The sample A06 areas of 1F9 monoclonal antibody at various concentrations.

**Figure 7 micromachines-13-00426-f007:**
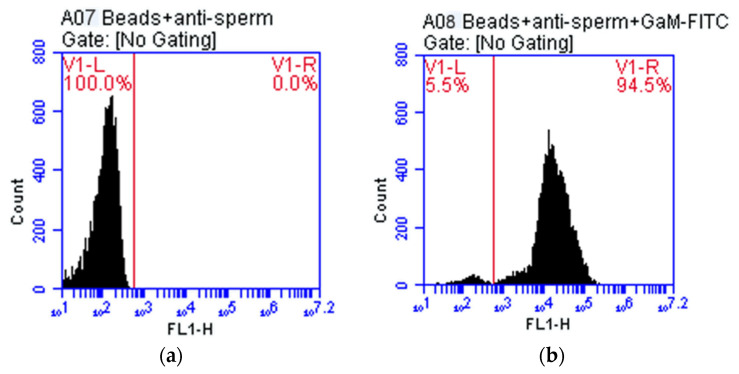
Physical characteristics of magnetic bead on monoclonal antibody 1F9 showing the relationship between the correlation between the Fluorescent Light Intensity (FL1-H) and the number of cells (Count) in the histogram (**a**) The magnetic bead on monoclonal antibody 1F9 (**b**) the Magnetic Bead on Monoclonal Antibody 1F9 stained with GαM-FITC.

**Figure 8 micromachines-13-00426-f008:**
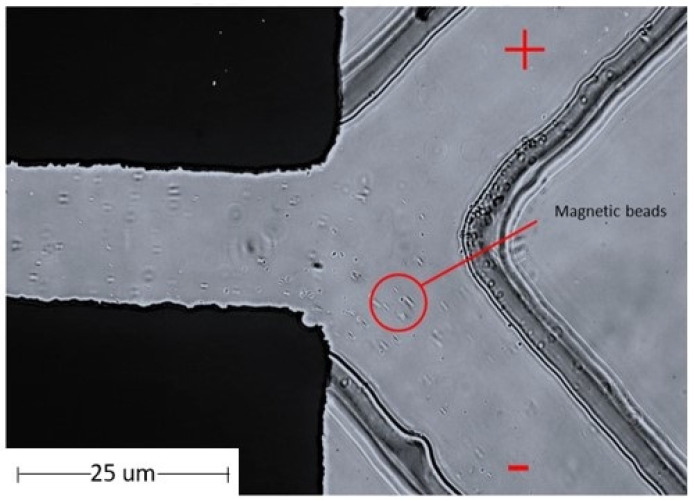
The flow of the magnetic particle in the microstructure flow channel with microfluidic chip.

**Figure 9 micromachines-13-00426-f009:**
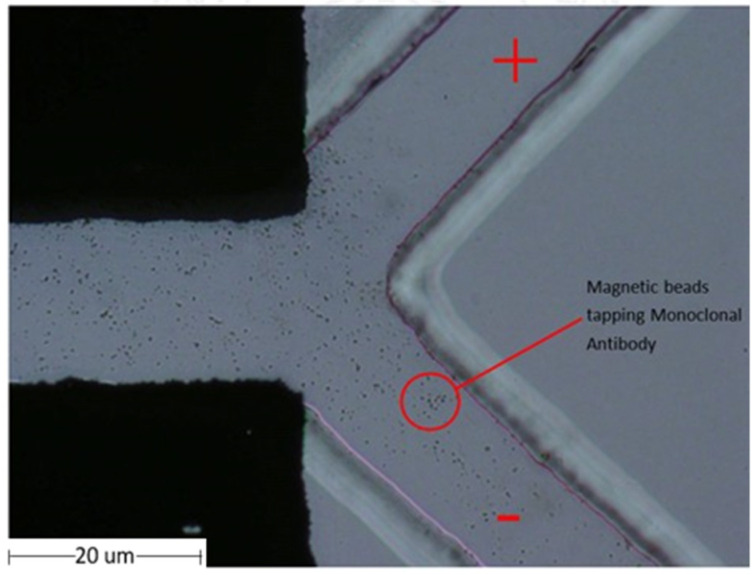
Magnetic particles flow on a monoclonal antibody in microstructure flow channel with microfluidic chip.

**Figure 10 micromachines-13-00426-f010:**
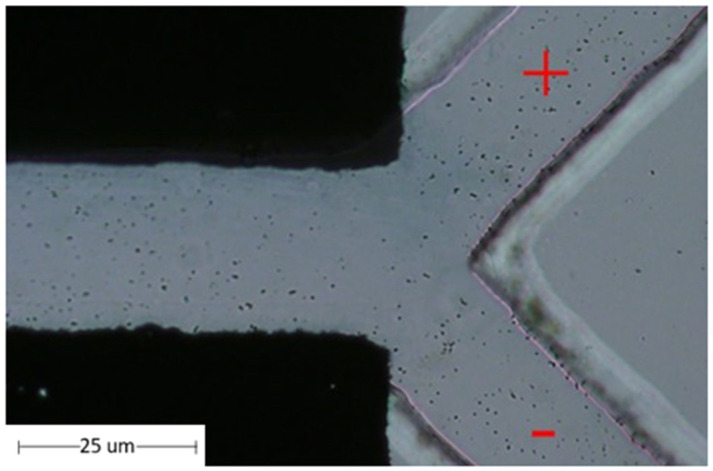
Flow of the magnetic particles on a monoclonal antibody and sperm in microstructure flow channel with microfluidic chip.

**Table 1 micromachines-13-00426-t001:** Conductivity data of MWCNTs for created in Microfluidic chip.

Material	MWCNTs	MWCNTs + PDMS 0.5 g:10 g	MWCNTs + PDMS 1 g:10 g
Voltage (V)	Current (A)	Current (A)	Current (A)
0.1	1.44 × 10^−3^	No Value	No Value
0.2	2.91 × 10^−3^	No Value	No Value
0.3	4.45 × 10^−3^	No Value	No Value
0.4	6.08 × 10^−3^	No Value	No Value
0.5	7.85 × 10^−3^	1.85 × 10^−9^	6.62 × 10^−8^

**Table 2 micromachines-13-00426-t002:** Parameter of sample A01–A06 was prepared for analyzed using flow cytometry.

Parameter	Preparation of Sample
A01	50 μL of FACS buffer and 50 μL of bull sperm
A02	1 μL of ST28A antibody, 50 μL of FACS buffer, and 50 μL of bull sperm.
A03	50 μL of 100 μg/mL MAb, with a final 1F9 MAb concentration of 50 μg/mL
A04	50 μL of 50 μg/mL MAb and 50 μL of bull sperm, with a final 1F9 MAb concentration of 25 μg/mL.
A05	50 µL of 25 µg/mL MAb and 50 µL cow sperm, with a final 1F9 MAb concentration of 12.5 µg/mL
A06	50 µL of 12.5 µg/mL MAb and 50 µL of cow sperm, with a final 1F9 MAb concentration of 6.25 µg/mL

**Table 3 micromachines-13-00426-t003:** Physical Characteristics of Magnetic Particle Beads on Monoclonal Antibody Stained with GαM-FITC (Plot 2).

Plot 2: A08 Beads + Anti-Sperm + GαM-FITC	Count	Events/µL	% of This Plot	% of All	Mean FL1-H	CV FL1-H
All	11,931	5966	100.00%	100.00%	25,833.83	177.21%
V1-L (1.0/528.0)	652	326	5.46%	5.46%	144.73	78.39%
V1-R (528.0/16,777,215.0)	11,279	5640	94.54%	94.54%	27,318.83	170.78%

**Table 4 micromachines-13-00426-t004:** Separation of sample magnetic particle beads with microfluidic chip and MWCNT-microfluidic chip.

Type of Electrode	Voltage (V)	Sorting of Magnetic Particle Beads in Microfluidic Capacity (%)
Microfluidic Thin film electron	2.5	100
MCNT-Microfluidic chip	2.5	100

**Table 5 micromachines-13-00426-t005:** Separation of the sample magnetic particles on monoclonal antibody in microstructure flow channel with microfluidic and MWCNT-microfluidic chip.

Type of Electrode	Voltage (V)	Sorting Magnetic Particles on Monoclonal Antibody Capacity (%)
Microfluidic Thin film electron	2.5	95.42
MCNT-Microfluidic chip	2.5	98.84

**Table 6 micromachines-13-00426-t006:** Efficiency results of X and Y sperm separation based on the measurement using real-time Polymerase chain reaction (PCR) techniques.

Type of Electrode	Sorting Sperm Y/X Capacity (%) without Magnetic Beads (Bottom)	Sorting Sperm Y/X Capacity (%) without Magnetic Beads (Top)	Sorting Antibody + Magnetic Beads + Sperm Y/X (Bottom)	Sorting Antibody + Magnetic Beads + Sperm Y/X (Top)
Thin film electrodes microfluidic chip	53.99/46.01	50.39/49.61	61.11/38.89	31.33/68.67
MCNTs electrodes microfluidic chip	74.62/25.38	51.29/48.71	80.12/19.88	28.56/71.44

## Data Availability

The data supporting reported results could be obtained by requesting.

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
