# Peer review of "Antibody-Conjugated Magnetic Beads for Sperm Sexing Using a Multi-Wall Carbon Nanotube Microfluidic Device"

_micromachines, 2022, doi:10.3390/mi13030426_

Round 1

Reviewer 1 Report

The authors presented a method for separating sperms using both electrostatic and magnetic forces. The aimed application is clear and the results are fine. However, the design choices are quite debatable and the quality of writing is low in terms of both language and logic. I would recommend an intensive revision to improve the quality of this paper.

Detailed comments:

  1. How is the magnetic field applied on chip and where exactly is the magnet placed? Also, wouldn't the particles accumulate on the side walls due to the application of the magnetic field and cause clogging of channels?
  2. The logic of the introduction part is quite twisted. I cannot understand the novelty of this proposed method. Why wouldn't one just choose to separate the sperms using magnetic forces only? Why is the electrostatic force and the microfluidic channels necessary? E.g., one can just replace the electrode and place a magnet there. Rationale on this point is pivotal to judge whether the use of microfluidic channel is novel or not. Please present a more clear and cohesive logic to justify your design choices.
  3. The two outlets are designed with same flow resistance, which can lead to unwanted flow of sperm X into the lower channel. Why not decrease the flow resistance for the upper channel so that less sperm X can flow into the lower channel?
  4. The authors wrote that the MWCNTs were pressed with a hydraulic machine and placed into the Ti/Au microelectrode. I could not understand this process, could the author provide some detailed illustration on this? Also, I would guess it more easy for the CNTs to be pressed into the PDMS part.... Anyway, I am quite confused on this fabrication step.
  5. Line 325 "width of 100", no unit.
  6. Abstract: the first two sentences were repeated twice.
  7. Please proof-read the manuscript carefully and have a through check on these simple misses/typos. I would recommend the authors use editing services to ensure the quality of it.
  8. Fig. 1a is not necessary to be presented here, it gives a false impression that the proposed system is Fig. 1a. Also, the magnetic separation is a quite common technique, no need to highlight it here. 

Reviewer 2 Report

In this manuscript, the author describes a separation method for sperm sexing in a microfluidic device. antibody conjugated magnetic beads are used to enhance the separation of Y sperm in an improved Y channel. The result demonstration is straight forward, and the author gives a detailed introduction of the research background and methods in the introduction session. The manuscript can be considered for publication if the following issue can be fixed:

  1. In the manuscript, the author covered a lot of different topics, including the device manufacturing process, antibody conjugated magnetic beads preparation, multi-wall carbon nanotube device, and sperm separation process. The main topic and claim of the manuscript are unclear. For each part mentioned in the manuscript, the author provides insufficient results to support the claim. The manuscript is bloated and is hard for the reader to follow. It is suggested that the author focuses on the main claim and present detailed results and discussion on the sperm separation process.
  2. Repeated sentences appear in the abstract, which need to be fixed.
  3. In the introduction session line88-110, the author has a detailed discussion of the motility difference of X and Y sperm in the electric and magnetic field. In the discussion session, the author should also discuss the influence of sperm motility to the separation process in the proposed experimental setup.
  4. In the last paragraph of the introduction, the author claimed that a permanent magnet is used to improve the separation quality as is described in Figure 1(b). However, the experiment process and results related to permanent magnet improved separation is not mentioned in the methods and results session. The author should add the experiment results and discussion of the influence of permanent magnets.
  5. In session 2.2, please add the dimension and density of magnetic particles.
  6. How does the conjugated magnetic beads affect Y-sperm motility without an electric field? 
  7. A flow chart could be helpful to describe the magnetic particle preparation process described shown in session 2.4. 
  8. It is suggested to use a table describing the test groups A01 - A06. 
  9. Session 2.6.1 described the lift off process for thin film electrode manufacturing process. While Figure 3 (a) demonstrated the metal etching process, which is inconsistent with text description. 
  10. Why lift-off method is used for Microfluidic chip electrode fabrication, and the etching method is used for MWCNT CHIP electrode fabrication on glass substrate?
  11.  Whether the author tried to separate magnetic beads conjugated sperm using a permanent magnet on the top of the device directly without electric field? It would be interesting to compare the separation efficiency difference between the magnetic separation and electrophoresis separation process.
  12. The author should add the description of the method to calculate separation efficiency of magnetic beads and sperm shown in Table 3 - 5.
  13. Did the author check the sperm survival rate and DNA damage after separation?
  14. Please list the distribution of X and Y sperm in two outlets of the Y-channel.
  15. In the MCNT-microfluidic chip, Y-sperm separation efficiency is 75% without magnetic beads conjugation and 80% with magnetic beads conjugation. From the viewpoint of economy and operation complexity, whether the magnetic beads conjugation process is necessary for the bovine sperm sexing process?

Reviewer 3 Report

This manuscript presents sperm sexing by antibody conjugated magnetic beads in a CNT microfluidic device. The idea is simple and seems effective. However, the introduction part needs to be reorganized. The presentation and scientific soundness are low. I would suggest the authors to address the following concerns.

  1. The introduction part needs to be re-organized. Some of the introduction part is redundant. For example, it is not necessary to introduce microfluidics and sorting by a whole paragraph 2. After reading four long paragraphs, I haven’t got what the authors tried to solve. I would recommend a short and precise introduction in the following logic. Background information about sperm sexing and its importance should be limited no more than one short paragraph, which is the 1st background paragraph. 2nd paragraph should focus on the current sorting techniques and their advantages and drawbacks. 3rd paragraph to expound the mechanism of your proposed sorting method and also a literature review summary for similar techniques, what other people has done so far. Then, the last paragraph to briefly summarize the proposed method and point out why this method been chosen for sperm sexing and why your system is better than others.
  2. Same for materials and methods part. For example, line 177-195, there’s no need for authors to introduce why choose carbon and CNT here. Just describe the materials and methods objectively. If this information is important, the authors can introduce in the introduction part.
  3. Lack of experiments to estimate the zeta potential and verify the surface charge
  4. Lack of experiment to verify the conjugation. This will make sure antibody is conjugated to magnetic beads.
  5. Figure 3c and 4c, please provide scale bars. Figure 3, please keep it consistent. a, b, and c or a., b., and c.?
  6. Just one suggestion about the fabrication process. Soft lithography and PDMS molding are quite standard processes in microfluidic device fabrication. The authors can think about illustrating the fabrication process in a simple way instead of explaining each step. Figure 3 and 4 are too crowded.
  7. Figure 5, what is the point of presenting device by UV lithography. Just present the method that used in the final device.
  8. Figure 6 and 7, why there is background in the figures?
  9. Figure 8, 10, and 11, please magnify to make the particles clearer and add scale bar.
  10. Figure 9, this figure should not be present in result part.
  11. A comparison to the published works should be added and discussed what's new and better in the proposed method.

Round 2

Reviewer 1 Report

I thank the authors for revising the manuscript accordingly. Most of the issues are addressed, except some minor checks are still required. After addressing these issues, I would be happy to recommend acceptance of this paper.

  1. line 539-552. Please be more specific on how to adjust the parameters of the microchannel and cite some papers which implemented this design strategy. Otherwise, the readers will just wonder how without any further clue.
  2. line 342-344. Are the illustrations in Fig. 5b correct? Why would the text "Developer" appear on the silicon substrate? Why are the replicated PDMS pattern and the photoresist pattern not fully complementary? I think the authors are providing various cross-sections of the device without indicating them correctly. Please be very careful to check them again.

Reviewer 2 Report

Most of the round 1 comments have been addressed properly and the manuscript is in good shape now. 

Additional comments:

  1. It would be good to include the permanent magnets separation result in this manuscript to improve the quality of this manuscript. 
  2. In the revised figure 1, the change of outlet sperm collection area leads to the change of channel resistance, and two branches do not have equal flow resistance.
  3. How does the electrical field effect X sperm distribution under different test conditions? Whether X sperm distribution is affected by the existence of Y sperm and Y sperm coated with magnetic beads?

Reviewer 3 Report

The authors have addressed my concerns. I would recommend acceptance of this manuscript.

Author Response

No comment from reviewer 3.